# Epigenetic Modifications Associated with Exposure to Endocrine Disrupting Chemicals in Patients with Gestational Diabetes Mellitus

**DOI:** 10.3390/ijms22094693

**Published:** 2021-04-29

**Authors:** Mateusz Kunysz, Olimpia Mora-Janiszewska, Dorota Darmochwał-Kolarz

**Affiliations:** Department of Obstetrics and Gynecology, Medical College, University of Rzeszow, 35-959 Rzeszow, Poland; olimpiajaniszewska@gmail.com (O.M.-J.); ddarmochwal@ur.edu.pl (D.D.-K.)

**Keywords:** EDCs, gestational diabetes mellitus, epigenetics

## Abstract

Gestational diabetes mellitus (GDM) remains a significant clinical and public health issue due to its increasing prevalence and the possibility for numerous short- and long-term complications. The growing incidence of GDM seems to coincide with the widespread use of endocrine disrupting chemicals (EDCs). The extensive production and common use of these substances in everyday life has resulted in constant exposure to harmful substances from the environment. That may result in epigenetic changes, which may manifest themselves also after many years and be passed on to future generations. It is important to consider the possible link between environmental exposure to endocrine disrupting chemicals (EDCs) during pregnancy, epigenetic mechanisms and an increased risk for developing gestational diabetes mellitus (GDM). This manuscript attempts to summarize data on epigenetic changes in pregnant women suffering from gestational diabetes in association with EDCs. There is a chance that epigenetic marks may serve as a tool for diagnostic, prognostic, and therapeutic measures.

## 1. Introduction

Diabetes is one of the most common metabolic diseases globally. The prevalence of pregestational and gestational diabetes continues to increase and is now at the upper end of previous estimates in Europe [1]. Gestational diabetes is detected in approximately 15% of pregnancies [2]. Although GDM usually resolves after delivery, it may have numerous, long-lasting health consequences, such as increased risk for type II diabetes mellitus (T2DM) and cardiovascular disease in the mother, and future metabolic and cardiovascular complications such as increased adiposity or even obesity, impaired glucose metabolism, hypertension, hyperlipidemia, and nonalcoholic fatty liver disease in the offspring, as well as preterm puberty, but only in girls [3,4,5,6,7]. The increasing incidence of GDM seems to coincide with the growing use of endocrine disrupting chemicals (EDCs). EDCs are a group of chemical substances that, due to their structural similarities with steroid hormones, interact with receptors for estrogens, androgens, and progesterone, thereby increasing the risk of endocrinopathy and diseases of civilization [8]. Exposure of pregnant women to EDCs might be a cause of the growing incidence of gestational diabetes, and as a consequence, increased risk of epigenetic “diabetogenic” and “obesogenic” changes in the offspring DNA, thereby escalating the risk of developing civilization diseases in subsequent generations. There are many examples in the literature that provide proofs of concept that EDCs should be treated as possible risk factors for poor fetal and maternal outcomes [9]. On the basis of existing evidence concerning the widespread effects of EDCs as well as the potential diabetogenic actions of particular EDCs, it is important to consider the possible association between environmental exposure to EDCs during pregnancy and an increased risk for developing gestational diabetes mellitus (GDM). Exposure of the maternal-fetal unit to EDCs may also, and perhaps more likely, result in epigenetic changes, which may manifest themselves immediately or after many years and be passed on to future generations. This sets in motion a vicious intergenerational cycle of metabolic diseases that influence the health of the human population [10]. In our review we attempted to emphasize that EDCs contribute to metabolic complications not only in affected mothers, but also in their offspring. In this way, pathologic changes can be transferred to subsequent generations, increasing the frequency of obesity, diabetes, and polycystic ovary syndrome, concerning even younger people. Given that some epigenetic changes are reversible, identified epigenetic marks may become important therapeutic targets, as well as potential biomarkers.

## 2. Patophysiological Aspects of Gestational Diabetes Mellitus

In early gestation, insulin sensitivity increases, promoting the uptake and storage of glucose to meet the energy demands of later pregnancy [11]. Increasing insulin resistance instead is one of the most important metabolic adaptations in the course of pregnancy, emerging at 14 weeks’ gestation and increasing up to two-fold by late pregnancy [12]. This process is caused by a rise in the concentration of pregnancy hormones—estrogen, progesterone, leptin, cortisol, placental lactogen, and placental growth hormone, which enable an adequate supply of glucose to the developing fetus. In most women, the increase in insulin requirements is compensated for by the growing insulin production. Evidence in animals suggests that this process results from hypertrophy and hyperplasia of pancreatic β-cells, as well as increased glucose-stimulated insulin secretion [13]. Maternal insulin sensitivity returns to pre-pregnancy levels within a few days after delivery [14].

Physiologic metabolic adaptations to pregnancy do not always function adequately during pregnancy, and both β-cell impairment (reduced β-cell mass, reduced β-cell number, β-cell dysfunction, or a mix of all three), and tissue insulin resistance are critical components of the pathophysiology of GDM. Additional factors contributing to the development of gestational diabetes have also been identified: adipose expandability, low-grade chronic inflammation, and oxidative stress [15,16,17]. Increased gut permeability is also thought to facilitate the movement of inflammatory mediators from the gut into the circulation, promoting systemic insulin resistance [18].

GDM is defined as glucose intolerance that is first diagnosed in pregnancy. Maternal hyperglycemia increases transplacental glucose transfer to the fetal circulation, resulting in overstimulation of the fetal pancreas. Physiologically insulin does not pass the placenta and the fetus begins to produce its own insulin around 9 weeks of age. Fetal hyperinsulinemia intensifies the metabolism and growth of the fetus (an overgrowth of muscle tissue, including the heart muscle, adipose tissue, and liver), increasing the demand for oxygen, especially in the last stage of pregnancy. As a consequence, fetuses from pregnancies complicated by GDM are more likely to suffer from intrauterine hypoxia and perinatal injuries resulting from excessive birth weight. The offspring of mothers with GDM may also have an increased risk for long-term sequelae. GDM increases the risk of maternal and fetal complications during pregnancy and long-lasting health consequences, such as amplifying the risk of T2DM and cardiovascular disease in the mother, and future metabolic and cardiovascular complications such as increased adiposity or even obesity [3,4,5,19], impaired glucose metabolism [6], hypertension [7], hyperlipidemia, and nonalcoholic fatty liver disease in the offspring, as well as preterm puberty, but only in girls [20,21,22,23]. GDM is associated with an enlarged risk for pre-eclampsia, birth complications such as shoulder dystocia and neonatal hypoglycemia, and an extended rate of cesarean section [24,25]. Mentioned GDM complications may represent intergenerational epigenetic inheritance, but embryonic exposure to an altered intrauterine environment also might cause an epigenetic transgenerational effect. It seems more and more likely that epigenetic factors have an important role in the complex interplay between genes and the environment. These interactions may result in the activation or deactivation of genes by epigenetic mechanisms, facilitating adaptation to environmental changes [26].

Extensive primary and secondary prophylaxis allows for early diagnosis of the disease, but a better screening strategy could result in diagnosis and treatment at a much earlier stage of pregnancy. Thus, there is strong a need to identify currently overlooked root causes of GDM and new and more accurate GDM risk biomarkers [27].

## 3. Overview of Epigenetics

A change in gene activity without a change in the nucleotide sequence is known as an epigenetic modification. Epigenetic modifications can be transmitted through cell division (mitotic inheritance) and through subsequent generations (meiotic inheritance) [28]. These modifications can be induced by the occurrence of some environmental factors influencing biologic systems, making them important pathogenic mechanisms in the development of complex multifactorial diseases. Intergenerational epigenetic inheritance is considered when direct environmental influence cannot be ruled out, and transgenerational epigenetic inheritance is defined as germ line-mediated inheritance of epigenetic information between generations in the absence of a direct stressor that leads to phenotypic variation. Exposure during pregnancy has a direct influence on the mother, fetus (intergenerational), and developing primordial germ cells of the growing fetus (transgenerational inheritance) [29,30], as seen on Figure 1. DNA methylation, histone modifications, noncoding RNA regulation, and chromatin remodeling are the main processes of epigenetic reprogramming, whose basic role in the milieu of the genome is to react to external and internal factors by dynamic, reversible changes in chromatin structure and gene expression [31]. Disturbances to this complementary process are believed to be the cause of a growing incidence of several multifactorial diseases, including GDM [32].

DNA methylation is the most frequently studied modification. Methylation turns off repressor elements and increases gene expression, while methylation at the promoter regions decreases gene expression. Moreover, increased expression or alternative splicing may be caused by methylation in the gene body [33]. It is mainly regulated through methylation of CpG islets. Histone modification is another type of epigenetic regulation that can influence chromatin packing and subsequently gene expression [34]. Yet another type of epigenetic regulation is micro RNA (miRNA), which is involved in the post-transcriptional regulation of gene expression. Micro RNA is a short non-coding RNA that can affect both the translation and stability of mRNA. Most often, miRNA binding to mRNA leads to translational inhibition or destabilization [35].

## 4. Epigenetic Regulations in Gestational Diabetes Mellitus

An aberrant intrauterine environment may induce changes in gene expression by epigenetic mechanisms, thereby affecting the development of the fetus and altering the offspring’s long-term risk for obesity and metabolic diseases such as diabetes in adulthood [36]. In utero, maternal glucose freely crosses the placenta, whereas maternal insulin does not, resulting in overstimulation of the fetal pancreas that is exposed to high glucose levels in GDM pregnancies. Increased need for insulin in the early stage of life is a likely trigger of epigenetic changes involving genes crucial for pancreatic development; β-cell development, differentiation, and function; peripheral glucose uptake; and insulin sensitivity. The transgenerational persistence of the insulin-resistant phenotype suggests that the epigenotype can be transmitted to the next generation [37,38,39].

Epigenetic modifications, although they do not change the genome as a whole, can be mitotically stable over time, causing long-term changes in gene expression [40]. The importance of epigenetic marks emerging at a very early stage of human development as potential modulators and predictors of future health and diseases was recently mentioned [40,41,42]. Periconceptional and intrauterine periods are crucial for fetal programming [43]. Epigenetic changes seem to take part in the complex interplay between genes and the environment that is related to insulin resistance, T2DM, and GDM. In fact, a growing number of studies have identified specific gene variants for susceptibility to GDM [44,45,46,47], as well as epigenetic alterations [40,41,42,48]. One of the largest groups among genes being investigated for their connection to diabetes are those linked to β-cell function and insulin secretion. The predisposition might be increased in the presence of transcription factor 7-like 2 (TCF7L2), hepatocyte nuclear factor 4 alpha (HNF4a) polymorphisms, and variants among glucokinase and glucokinase regulatory protein genes. However, the presented inherited genetic variants may be under the influence of the mentioned epigenetic mechanism. Some of the most significant studies investigating the epigenetic background of GDM focused on mechanisms concerning gene silencing or augmentation are presented below. The ATP-binding cassette transporter A1 is another gene whose demethylation reduces high-density lipoprotein concentrations and raises glucose levels in OGTT [49].

In the EPOCH study, Yang et al. [50] divided patients into two groups: offspring of GDM and non-GDM mothers. Samples of cord and peripheral blood were obtained to identify GDM-associated DNA methylation areas and assess their possible association with child metabolic outcomes. The researchers identified significant changes in 51 areas of the genome and demonstrated that the methylation of five genes is linked to GDM. They demonstrated that methylation at the differentially methylated position in SH3PXD2A was significantly positively correlated with adiposity-related outcomes: BMI; waist circumference; triceps, suprailiac, and subscapular skinfold thickness; subcutaneous adipose tissue quantity; and leptin levels. DNA methylation in E2F6 was also associated with fasting insulin and the homeostatic model assessment for insulin resistance (Homa-IR). According to the researchers, these findings suggest that DNA methylation is affected by GDM exposure in utero and epigenetic changes may represent a significant link between this exposure and childhood obesity. SH3PXD2A hypomethylation was associated with GDM exposure in cord blood at birth [51], and with T2DM in pancreatic islets [52]. Furthermore, a CpG site within its homologue SH3PXD2B is also hypomethylated in severe childhood obesity [53]. Although the biologic function of the SH3PXD2A involvement in diabetes and obesity remains unclear, these multiple studies strongly suggest that it plays a crucial role in the molecular basis of metabolic disorders. As reported by Weng et al., 37 methylated CpGs (representing 20 genes) between the GDM and healthy groups were identified and showed potential as clinical biomarkers for GDM, suggesting it has epigenetic effects on genes that are preferentially involved in the Type I diabetes mellitus pathway, immune MHC-related pathways and neuron development-related pathways. The analysis was performed on umbilical cord blood [54]. Ruchat et al. demonstrated that differentially methylated genes identified in the placenta and in cord blood were also correlated with newborn weight [48]. A recent study conducted by Chen et al. identified differentially methylated CpGs in 39 genomic regions influenced by in utero exposure to GDM in offspring’s peripheral blood [55]. Methylation at three sites was also nominally associated with insulin secretion, while a fourth site was associated with a future risk of T2DM. Complex interactions between genetic (probably polygenic) susceptibility, unfavorable fetal surroundings, and the environmental impact of chemicals such as EDCs may lead to activation or deactivation of genes by epigenetic mechanisms, enabling adaptation (to some extent) to miscellaneous environmental situations, but sometimes bringing about the development of various disorders.

Epigenetic modifications have become a probable link between the in utero exposure to a diabetic environment and poor outcomes of the offspring. The association of GDM and diabetes during pregnancy with the epigenetic changes detected in in utero-exposed children has been confirmed by many others [40,41,42,55,56,57].

Wu et al. found significant differences in methylation patterns among women who developed GDM compared with those who did not [58]. Moreover, another study conducted by Reichetzeder et al. revealed that patients who develop GDM had higher levels of DNA methylation in placental tissues [59]. According to Michalczyk et al., the histone H3K27 and H3K4 demethylation levels are correlated with GDM progression to T2DM. The percentage of H3K27 and H3K4 methylation was lower in women who develop T2DM later in life than in women who are unaffected by T2DM post-GDM [60]. Another study performed by Wander et al. showed that miR-155-5p and miR-21-3p plasma levels in early pregnancy are associated with a higher risk of GDM. The miRNAs miR-21-3p and miR-210-3p were also linked to GDM, but only in overweight women. Moreover, pregnant women with male fetuses had a higher risk of GDM because of interferences in the plasma levels of miR-29a-3p, miR-223-3p, miR146b-5p, and miR-517-5p [61]. According to Zhao et al., pregnant women with considerably decreased expression levels of miRNAs miR-29a, miR-132, and miR-222 later developed GDM. miR-29a is involved in glucose metabolism, but the roles of miR-222 and miR-132 have yet to be determined [62].

Wu et al. [58] reported DNA methylation modifications in the blood of gravid women even before GDM was detected. They recognized a set of diversely methylated genes shared by the blood, umbilical cord, and placenta: retinol dehydrogenase 12, hook microtubule tethering protein 2, phosphoinositide-3-kinase regulatory subunit 5, constitutive photomorphogenic homolog subunit 8, coiled-coil domain containing 124, 3- hydroxyanthranilate 3,4-dioxygenase, and chromosome 5 open-reading frame 34.

Cardenas et al. [63] carried out an epigenome-wide association study on samples of placenta and plasma glucose, matching them up with 2 h post-OGTT plasma glucose levels. They discovered that plasma glucose at 2 h OGTT positively correlates with reduced DNA methylation of four CpG sites within the phosphodiesterase 4b gene. Moreover, three other CpG sites in the TNFRSF1B, LDLR, and BLM genes were found to be differentially methylated in association with maternal glucose.

Martinez-Ibarra et al. [64] explored the connection between GDM, EDCs, and miRNA. They observed higher levels of miR-9-5p, miR-29a-3p, and miR-330-3p in sera of patients with GDM compared to non-diabetic subjects. Moreover, according to Li et al., expression levels of miR-9-5p were significantly decreased in placental villous tissues and cytotrophoblast of GDM patients [65]. miR-9-5p directly targets hexokinase-2 (HK-2) and affects its expression. HK-2 was upregulated in both placental villous tissues and cytotrophoblasts from GDM patients compared with healthy women. Zhang et al. demonstrated that miR-9-5p significantly reduces the expression of glucose transporter 1 and glycolytic enzymes (HK-2, phosphofructokinase, and lactate dehydrogenase) [66]. Another study conducted by Kong et al. showed that overexpression of miR-9-5p in the serum of newly diagnosed T2DM patients was also found to reduce the expression of glucose transporter 1, hexokinase-2 (HK-2), phosphofructokinase, and lactate dehydrogenase, compared with a control group [67]. Sebastiani et al. also presented that miR-29a-3p modifies the expression of various genes involved in the insulin signaling pathway, such as HK2, and negatively regulates fatty acid oxidation through peroxisome proliferator-activated receptor γ coactivator 1-α expression. The expression levels of miR-330-3p, which is related to the proliferation and differentiation of β-cells and insulin secretion, are were high [68]. Table 1 summarizes some of the studies.

## 5. General Aspects of Endocrine Disrupting Chemicals

Endocrine disrupting chemicals (EDCs) are a group of chemical substances that, due to their structural similarities with steroid hormones, interact with receptors for estrogens, androgens, and progesterone, thereby increasing the risk of endocrinopathy and diseases of civilization [8]. Currently, the EDC list includes hundreds of chemical compounds and it is constantly growing. As reported by the European Union, from a total of 564 chemicals that have been suggested by various organizations in published papers or reports as being suspected EDCs, 147 are considered likely to be either persistent in the environment or produced at high volumes. A reliable estimation of the number of EDCs is practically impossible, however, due to the constant introduction of new chemical substances and the formation of their active derivatives.

EDCs comprise a heterogeneous group of synthetic and natural chemical compounds, most of which have phenol groups in their structure, conferring their affinity to the steroid hormone receptors: estrogens, progesterone, and androgens. EDCs have agonistic or antagonistic effects on nuclear receptors, which are their major targets [69]. Among them, the greatest health concern is linked with plasticizers (phthalates and bisphenol A (BPA) and its derivative bisphenol S) and pesticides: DDT (dichlorodiphenyltrichloroethane), chlorpyrifos, methoxychlor, fungicides (vinclozolin), herbicides, polychlorinated biphenyls, brominated flame retardants, and per- and poly-fluoroalkyl substances, the health effects of which are described in numerous publications [70]. These compounds enter the body by ingestion, inhalation, and absorption through the skin. EDCs were detected in body fluids: blood serum, saliva, tears, urine, stool, milk, semen, amniotic fluid, placenta, meconium, and fat tissue in tested populations around the world [71,72,73,74,75,76,77]. Leaching into the soil and groundwater, they enter into the food chain by accumulating in fish, animals, and plants. Some consumer products such as household chemicals, fabrics enriched with flame retardants, cosmetics, lotions, products with fragrance, and anti-bacterial soaps contain EDCs or are packaged in containers that can leach EDCs. Processed food can accumulate traces of EDCs that leach out of materials used in manufacturing and storage. EDCs such as lead, flame retardants, and polychlorinated biphenyls from furniture pollute household dust. Lipophilic EDCs can remain in the human body for years and be secreted from adipocytes, and then, after binding with the appropriate receptors, modify the hormonal response, whereas others are removed from the organism relatively quickly.

Daily, constant exposure to EDC mixtures in concentrations even below the established threshold tolerated by the human body for individual substances can significantly increase the risk of developing hormonal and metabolic disorders, such as obesity, diabetes, polycystic ovary syndrome, and cardiovascular disease, as well as malformations, infertility, and hormone-dependent cancers in both women and men [78,79,80,81,82,83,84,85,86,87].

Civilizational development and the growing demand for new chemical substances increases our exposure to endocrine active chemicals. The widespread production and common use of these substances in everyday life has resulted in constant exposure to harmful substances from the environment, including electronic equipment, furniture, paints, floors, and toys. Additional daily exposure occurs through the release of these substances from such commonly used items as food packaging, bottled drinks, cosmetics, receipts, clothes, food, contact lenses, and dental seals [88,89,90]. EDCs are absorbed through the skin, as well as by the consumption of contaminated food and breathing contaminated air [91]. Some EDCs may be even more prevalent in newborns and children than in adults in connection with greater consumption of specific foods and water. Additionally, ventilation rates, intestinal absorption, surface area to volume ratios, and hand-to-mouth activity are higher during infancy and puberty [92]. Breastfeeding is also associated with greater infants’ exposure to EDCs [93].

The effect of EDCs on adult organisms is different from that on developing organisms. In adult organisms, high doses of EDCs are needed for an effect, and the effect disappears when the exposure is discontinued. By contrast, in developing organisms, exposure to EDCs may often produce long-lasting effects. One possible reason that fetal exposure leads to more detrimental effects than adult exposure is the absence of adequate defense and detoxification mechanisms in prenatal life [94]. Compared with adults, the developing fetus has lower levels of cytochrome P450 enzymes, which are used by the body to metabolize environmental pharmaceuticals and chemicals [95,96].

After the identification of EDCs in the bodies of the pregnant mother and fetus [97,98,99,100], studies of their effects on fetal development and future health were initiated [101]. Several studies using model systems have demonstrated that external environmental stimuli such as EDCs can induce epigenetic mutations during gametogenesis, embryogenesis, and fetal development [31,102,103].

The pre- and perinatal period, infancy, childhood, and puberty are among the so-called critical periods of development during which the susceptibility to hormonal disorders induced by chemical stimuli is particularly high. A hostile intrauterine environment related to an adverse maternal lifestyle may be a causative factor of several metabolic adult-onset conditions, such as metabolic syndrome, type 2 diabetes mellitus (T2DM), and cardiovascular disease. Exposure of the fetus to EDCs may also, and perhaps more likely, result in epigenetic changes, which may manifest themselves after many years and be passed on to future generations. This sets in motion a vicious intergenerational cycle of metabolic diseases that influence the health of the human population [94,104,105,106].

There is a growing body of evidence that EDCs target several pathophysiologic features of GDM, such as weight gain, insulin resistance, and pancreatic β-cell function [84,107,108]. EDCs influence the activity of peroxisome proliferator-activated receptors, which are crucially involved in glucose metabolism and energy homeostasis. Regarding T2DM, animal studies indicate that some EDCs directly affect cells in the pancreas, adipocytes, and liver, and induce insulin resistance and hyperinsulinemia. These can also be associated with modified levels of adiponectin and leptin, often in the absence of obesity. This diabetogenic activity enhances the risk for cardiovascular diseases, and hyperinsulinemia augments the probability of diet-induced obesity. Exposure of pregnant women to EDCs can be a cause of the growing incidence of gestational diabetes, and as a consequence, increased risk of epigenetic “diabetogenic” and “obesogenic” changes in the offspring DNA, thereby increasing the risk of developing civilization diseases in subsequent generations. Many EDCs have been linked to metabolic effects in cell-based, animal, and epidemiologic studies.

One the most investigated EDCs in relation to its effects on the feto-maternal compartment is BPA (bisphenol A). BPA has been used worldwide since the 1960s to produce polycarbonate plastics (hard, transparent, with high thermal and mechanical resistance), but it is also used in epoxy resins and thermal paper. BPA can be found in everyday items such as compact discs and digital video discs, electronic equipment, and toys. Exposure already occurs in the prenatal period, then continues during lactation and through food consumed by the newborn. It is commonly detected in human blood and urine; regardless of age or sex, BPA enters the human body through the respiratory, digestive, and transdermal routes [109].

BPA is endocrine-active. Even small amounts of BPA may be toxic to the endocrine system [110]. The best known and documented BPA-related disorders are: increased likelihood of breast, uterine, and prostate cancer; impaired fertility, premature births, and miscarriages; and in vitro fertilization failure [111,112,113,114,115,116,117]. Exposure to BPA may also increase the risk of developing diabetes and obesity [118,119,120]. BPA exerts an estrogenic effect by activating estrogen receptors α and β. These receptors are ubiquitous in tissues and are involved in the regulation of blood glucose levels [121].

Phthalates are diesters of 1,2-benzendicarboxylic acid. They are widely used primarily in PVC, as general-purpose plasticizers in polymers. Day-to-day products containing phthalates are wall coverings, sealants, cables, floorings, coatings, adhesive paints, packaging materials, toys, roofings lacquers, and clothing [122].

Their absorption occurs through inhalation, ingestion or dermal absorption. Moreover, phthalates can cross the placenta, resulting in exposure to the fetus [123,124].

Parabens are another group of chemicals with endocrine activity. Chemically, they are esters of p-hydroxybenzoic acid with alkyl substituents ranging from methyl to butyl or benzyl groups. The most frequently applied are methylparaben (MeP) and propylparaben (PrP), ethylparaben (EtP), butylparaben (BtP), and benzylparaben (BenzylP). They are generally used as powerful antimicrobial substances and preservatives involved mainly in personal care products and pharmaceutic products, but also in a number of industrial products and foodstuffs. Parabens can be absorbed through ingestion, inhalation, and the skin. Parabens may cumulate in the body as a result of day-to-day use of paraben-containing products [125,126,127,128,129].

## 6. Association between Endocrine Disrupting Chemicals and Gestational Diabetes Mellitus

BPA may increase the risk of diabetes by altering the function of pancreatic β-cells and the action of insulin in target tissues. BPA disrupts pancreatic cell function, which leads to a failure of compensatory mechanisms and the development of hyperglycemia [130,131,132]. Wei et al. [133] showed that BPA promotes disturbances in insulin secretion in pancreatic cells that are associated with inhibition of pancreatic and duodenal homeobox 1 and decreased expression of miR-338.

Martinez-Ibarra et al. found high unadjusted mono(2-ethylhexyl) phthalate (MEHP) levels and high creatinine-adjusted BPA levels in non-diabetics compared to women with GDM. These findings could be explained by the similarities in the molecular structures of BPA and 17β-estradiol. They found no correlation, however, between MEHP and GDM.

Moreover, their results showed a significant positive correlation between adjusted urinary levels of mono-benzyl phthalate and expression levels of miR-16-5p in non-diabetic women. Expression levels of miR-29a-3p negatively correlated with non-adjusted and adjusted urinary mono-n-butyl phthalate concentrations and unadjusted urinary mono isobutyl phthalate concentrations, and positively correlated with adjusted urinary MEHP concentrations. Unadjusted urinary BPA concentrations correlated with pre-gestational BMI in women without GDM, but not in those with GDM.

Schaffer et al. [134] found that MEP is significantly associated with increased odds of developing GDM. Moreover, they detected a positive association between mono-n-butyl phthalate and impaired glucose tolerance, and between both mono-carboxy-iso-octyl phthalate and blood glucose. By contrast, mono-(3-carboxypropyl) phthalate was inversely associated with GDM. Li et al. [135] showed that among overweight/obese pregnant women, who represent a subgroup more prone to GDM, moderately higher levels of propylparaben and total estrogenic activity of parabens were significantly associated with increasing GDM prevalence.

Moreover, a study conducted by Bellavia et al. [136] found that first trimester butylparaben and propylparaben urinary concentrations are associated with glucose levels in a cohort of pregnant women at high risk of GDM, even after adjusting for potential confounders.

In a recent study, Hou et al. [137] demonstrated that exposure to 2-tert-octylphenol (2-t-OP) is associated with a higher risk of GDM. However, higher NP exposure is associated with lower GDM risk.

On the other hand, some studies have demonstrated no adverse effects in humans. A recent meta-analysis of eight studies revealed no noteworthy association between BPA exposure and birth weight [138]. Moreover, no correlation between prenatal and early-life levels of BPA and childhood adiposity was detected [123,139]. Furthermore, there was no evidence for an association among urinary BPA concentrations and in vitro fertilization outcomes [140]. Table 2 summarizes some of the studies.

## 7. Discussion

As presented above, several studies have demonstrated negative effects of EDCs on glucose homeostasis in pregnant women, some leading to metabolic changes in the offspring. Unfortunately, however, many of the findings are inconsistent. The limited evidence and inconsistent results do not allow us to draw a firm conclusion. The inconsistencies in the literature may be due to differences in maternal age, BMI, the set of probes, race and ethnicity, socioeconomic status, and educational background of the study participants. Among the three pregnancy terms, the middle term seems to be the most metabolically susceptible to external influences, and therefore, the timing of sample collection must be taken into consideration.

In addition, urinary levels of EDCs might not reflect levels in the placenta or fetus, which are believed to be affected by tissue accumulation, subsequent release, and clearance. Many studies have not taken into account factors such as maternal pre-pregnancy diseases, gestational maternal weight gain, parity, hemodilution of plasma samples, dietary choices, and medications, all of which can affect the maternal environment and offspring health. Exposures to various mixtures of substances also differ across human beings, and the need for costly diagnostic technologies often limits studies of the thousands of natural and synthetic compounds with endocrine effects.

Despite these constraints, analysis of the available literature provides proofs of concept that EDCs should be treated as possible risk factors for poor fetal and maternal outcomes. Some studies provide important clues to help spot the consequences of EDCs more rapidly—mainly disorders with extended latency phases.

In general, the evidence analyzed in this essay indicates that EDCs contribute to metabolic complications not only in affected mothers, but also in their offspring. In this way, pathologic changes can be transferred to subsequent generations. These assumptions are supported by the constantly increasing frequency of obesity, diabetes, and polycystic ovary syndrome, affecting even younger people. Unfortunately, there is a lack of studies analyzing mixtures of many chemicals.

Because adverse effects on the maternal-fetal compartment are highly probable, implementation of preventive measures is crucial. Education of medical staff and the general population, particularly pregnant women, putting pressure on governments and politicians to reduce plastic production to the necessary minimum, and setting appropriate methods for the purification of industrial and municipal sewage are actions that will be beneficial for human health. There is a pressing need to minimize the occurrence of these illnesses, which can be managed by studies matching exposure to EDCs with future outcomes. The expanding evidence regarding these environmental contributors to non-communicable diseases suggests that synthetic chemicals are often ignored.

## 8. Conclusions

A growing body of evidence has shown that the increasing incidence and long-term metabolic consequences of GDM may be mediated by epigenetic modifications. Epigenetic marks found in GDM mothers and their offspring might be considered as future targets for potential management or early diagnostic predictors of some metabolic diseases and even tools for personalized preventive healthcare strategies. Despite inconsistent results of some studies on EDCs, it is highly probable that substances with proven negative health effects also have a detrimental impact on the maternal-fetal unit. There is a need for further multi-directional interventional and longitudinal research studies on epigenetic modifications being induced by environmental pollutants such as EDCs. An increasing number of large-scale genome-wide researches might be able to provide a large amount of essential information. Bearing in mind that these studies are complex and expensive, while waiting for their results, it is worth trying to modify the probable risk factors. Concerning that the periconceptual period, intrauterine development, and early postnatal lifetime are the most crucial for programming future human health, it seems to be valuable to undertake education, especially among women of reproductive age and in particular among pregnant women. Because epigenetic changes are modifiable, it is a chance for a reduction of intergenerational inheritance of metabolic traits and reversing the unfavorable growth trend of metabolic diseases such as diabetes, obesity, and PCOS and their long-lasting consequences.

## Figures and Tables

**Figure 1 ijms-22-04693-f001:**
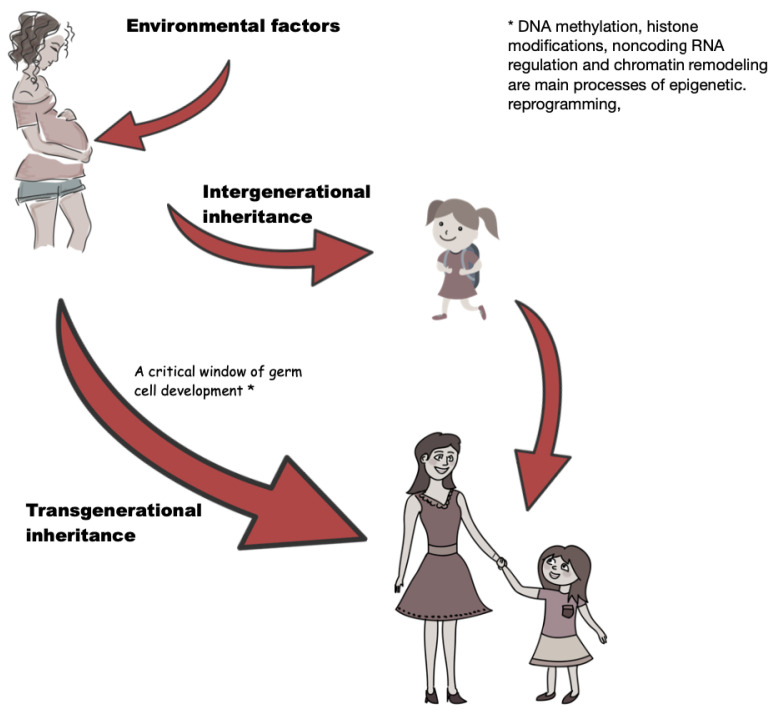
Exposure during pregnancy means direct influence on mother and fetus (intergenerational) and developing primordial germ cells of growing fetus (transgenerational inheritance).

**Table 1 ijms-22-04693-t001:** Epigenetic studies on gestational diabetes mellitus.

Study	Effect
Weng et al. [54]	Thirty-seven methylated CpGs linked to GDM
Yang et al. [50]	Significant changes in 51 areas of the genome and demonstration that the methylation of 5 genes is linked to GDM
Finer et al. [51]	SH3PXD2A hypomethylation is associated with GDM exposure in cord blood at birth
Chen et al. [55]	Differentially methylated CpGs in 39 genomic regions influenced by in utero exposure to GDM in offspring peripheral blood
Reichetzeder et al. [59]	Higher levels of DNA methylation in placental tissues in patients with gestational diabetes mellitus
Michalczyk et al. [60]	Histone H3K27 and H3K4 demethylation levels are correlated with GDM progression to T2DM.
Zhao et al. [62]	Decreased expression levels of miRNAs miR-29a, miR-132, and miR-222 associated with later development of GDM
Wander et al. [61]	miR-155-5p and miR-21-3p plasma levels in early pregnancy are associated with a higher risk for GDM
Wu et al. [58]	Differentially methylated genes shared by the blood, umbilical cord, and placenta
Cardenas et al. [63]	Three CpG sites in the TNFRSF1B, LDLR, and BLM genes are differentially methylated in association with maternal glucose
	Plasma glucose at 2 h OGTT positively correlates with reduced DNA methylation of 4 CpG sites within the phosphodiesterase 4b gene
Sebastiani et al. [68]	Increased expression levels of miR-330-3p
Li et al. [65]	miR-9-5p were significantly decreased in placental villous tissues and cytotrophoblast of GDM patients
Martinez-Ibarra et al. [64]	Higher levels of miR-9-5p, miR-29a-3p, and miR-330-3p in sera of patients with GDM compared to non-diabetic subjects

**Table 2 ijms-22-04693-t002:** Studies on the epigentic effect on patients with gestational diabetes mellitus.

Study	EDCs Exposure	Effect
Hou et al. [137]	Urinary 2-t-OP	Higher risk of GDM
	Urinary NP	Lower GDM risk
	Urinary BPA	No significant association with risk of GDM
Martínez-Ibarra et al. [64]	Unadjusted urinary MiBP concentration	Decreased miR-29a-3p expression levels
	Urinary MBP concentrations	Decreased miR-29a-3p expression levels
	Urinary MEHP concentration	Increased miR-29a-3p expression levels
	Urinary MBzP	Increased miR-16-5p expression levels
Shaffer et al. [134]	Urinary MCPP	Inversely associated with GDM
	Urinary MCOP	Increased blood glucose
	Urinary MBP	Increased impaired glucose tolerance
	Urinary MEP	Increased blood glucose
Li et al. [135]	Urinary propylparaben	Increased GDM prevalence in overweight/obese pregnant women
Bellavia et al. [136]	Urinary butylparaben and propylparaben	Increased glucose levels
Hu et al. [138]	Urinary BPA	N/C birth weight
Braun et al. [139]	Urinary BPA	N/C BMI at 2–5 but accelerated growth
Buckley et al. [123]	Urinary phenols	N/C influence on the development of childhood adiposity
Mínguez-Alarcón et al. [140]	Urinary BPA	N/C IVF outcomes

## Data Availability

Not applicable.

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
