# Peer review of "Epigenetic Modifications Associated with Exposure to Endocrine Disrupting Chemicals in Patients with Gestational Diabetes Mellitus"

_ijms, 2021, doi:10.3390/ijms22094693_

Round 1
Reviewer 1 Report
Kunysz et al review on the role of EDC on epigenetic modifications during pregnancy and gestational diabetes is a very important topic. The manuscript however did not focus on this topic. It started broadly with highly general descriptions of ETCs and with industrial aspects of ETC, not focusing on the topic itself. The manuscript has thrown a lot of mixing information in an unorganized manner; multiple statements are general and/or were not supported with literature. The authors mixed information from clinical and animal research and made their own unsubstantiated conclusions. The manuscript requires significant improvement before being accepted for publication.
Author Response
Dear Madam, Dear Sir,
thank you for giving us the opportunity to submit a revised draft of the manuscript “Epigenetic Modifications Associated with Exposure to Endocrine Disrupting Chemicals in Patients with Gestational Diabetes Mellitus” for publication in the International Journal of Molecular Sciences. We appreciate the time and effort that you dedicated to providing feedback on our manuscript and are grateful for the insightful comments on and valuable improvements to our paper.
We attempted to incorporate Your suggestions. We reduced information about EDCs and their industrial aspects. We tried to reorganize text in order to make it more readable and to emphasize what was the purpose and topic of the manuscript.
We managed to systematize researches and put them in the table. We agree that there are some limitation of the review but it is due to restricted data from literature.
Reviewer 2 Report
The present paper aims to summarise epigenetic changes in pregnant women with
gestational diabetes.
A few changes are needed, as follows:
Please explain every abbreviation before using it, starting with EDCs in the abstract.
Page 3, lines 92-99: Please provide references!
Conclusions contain many aspects which belong to Discussion. Please include a section which emphasizes and explains contradictory results.
Please include also future research directions in this area!
A table would allow a better systematization of studies on this topic.
Please emphasize in Conclusions the implications of Endocrine Disrupting Chemicals for clinical practice.
Author Response
Dear Madam, Dear Sir,
we appreciate the positive feedback from You. Thank you very much for your valuable suggestions. Thanks to them, we tried to make an improvement in our research paper. As suggested, we have explained abbreviations and added table of them. We have provided missing references. We attempted to improve organisation of the manuscript and added new section such as discussion. We did our best to emphasize implications of the EDCs and include future research directions in this area, mostly in conclusions. However we did not create another section for contradictory results because of limited quantity of researches. We introduced tables for better systematization. We are deeply open for every kind of comment.
Round 2
Reviewer 1 Report
The manuscript has been significantly improved and is acceptable for publication.